

# Differing responses of the QBO to $SO_2$ injections in two global models

Ulrike Niemeier[1], Jadwiga H. Richter[2], and Simone Tilmes[2]

[1]Max Planck Institute for Meteorology, Bundesstr. 53, 20146 Hamburg, Germany
[2]National Center for Atmospheric Research, Boulder, CO, USA

**Correspondence:** Ulrike Niemeier (ulrike.niemeier@mpimet.mpg.de)

**Abstract.** Artificial injections of sulfur dioxide ($SO_2$) into the stratosphere show in several model studies an impact on stratospheric dynamics. The quasi-biennial oscillation (QBO) has been shown to slow down or even vanish, under higher $SO_2$ injections in the equatorial region. But the impact is only qualitatively, but not quantitatively consistent across the different studies using different numerical models. The aim of this study is to understand the reasons behind the differences in the

QBO response to $SO_2$ injections between two general circulation models, the Whole Atmosphere Community Climate Model (WACCM-110L) and MAECHAM5-HAM. We show that the response of the QBO to injections with the same $SO_2$ injection rate is very different in the two models, but similar when a similar stratospheric heating rate is induced by $SO_2$ injections of different amounts. The reason for the different response of the QBO corresponding to the same injection rate is very different vertical advection in the two models, even in the control simulation. The stronger vertical advection in WACCM results in a

higher aerosol burden and stronger heating of the aerosols, and, consequently in a vanishing QBO at lower injection rate than in simulations with MAECHAM5-HAM.

## 1  Introduction

Recent model intercomparison studies of sulfate evolution and transport after volcanic eruptions and after artificial injections

of $SO_2$ into the stratosphere reveal substantial differences between model results. The lifetime of the aerosols after a simulated Tambora eruption differs by several months and the aerosol optical depth (AOD) shows different maximum values and decay rates (Zanchettin et al., 2016; Marshall et al., 2018). Similar differences in response are also found in Climate Engineering (CE) studies, in which $SO_2$ is continuously injected into the stratosphere over a period of many years. Niemeier and Tilmes (2017) show a wide range of radiative forcing values resulting from the same sulfur injection rate but in different models. Radiative

forcing results of the two models compared in Kleinschmitt et al. (2017) are closer but vary still by e.g. 0.5 $Wm^{-2}$ for an injection rate of 10 $Tg(S)yr^{-1}$. Kleinschmitt et al. (2017) assumed differences in aerosol heating and consequent stronger vertical advection as a reason for the differences.





Several models show that the artificial injection of $SO_2$ into the tropical stratosphere over many years impacts stratospheric dynamics. The quasi-biennial oscillation (QBO) is the primary mode of variability in the tropical stratosphere, characterized by downward propagating easterly and westerly shear zones. The QBO affects lower troposphere temperature and constituent concentrations, as well as affects transport of constituents out of the tropics (Baldwin et al., 2001; Punge et al., 2009; Shuck-burgh et al., 2001). The QBO slows down, or even vanishes, under higher $SO_2$ injections in the equatorial region (Aquila et al., 2014; Niemeier and Schmidt, 2017; Richter et al., 2017; Jones et al., 2016). Aquila et al. (2014) showed that the cause of this dynamical change in the tropical stratosphere are changes in temperature resulting from the radiative heating of the aerosols. Sulfate scatters solar (short wave, SW) radiation, which causes the earth surface to cool and absorbs radiation within the SW spectrum in the near infrared, as well as terrestrial (long wave, LW) radiation. This absorption causes the sulfate layer in the stratosphere to warm. Timmreck et al. (1999) and Aquila et al. (2012) have shown the importance of this radiative heating for the transport of sulfate after a volcanic eruption.

The heated sulfate layer is the main driver of the changes in the tropical stratospheric circulation and the QBO. The disturbed thermal wind balance results in an increased zonal westerly wind component (Andrews et al., 1987). Additionally, the heating increases the vertical advection as given in the residual vertical velocity, $\omega^*$, either direct, by changing the density of the air, or indirect by changing the propagation of waves in the steady state. Dissipating waves deposit their energy in the stratosphere. Therefore, changing temperature and temperature gradients with artificial sulfur injections at the equator changes stratospheric dynamics and tracer transport. Additionally, a stronger $\omega^*$ inhibits the downward propagation of QBO shear zones, resulting in a lengthening or total loss of an oscillation in the presence of larger $SO_2$ injections (Aquila et al., 2014; Niemeier and Schmidt, 2017; Richter et al., 2017). Changes in the QBO resulting from $SO_2$ injections subsequently have consequences for aerosol transport, due to the strong westerly jet in the lower stratosphere. A tropical westerly jet results in a stronger equatorward meridional wind component toward the center of the jet (Plumb, 1996) and, together with the enhanced vertical advection, an enhanced tropical confinement of the aerosols (Niemeier and Schmidt, 2017). To decrease the impact on the QBO other injection areas might be favorable. Richter et al. (2017) showed that the QBO period decreases when $SO_2$ injections are placed at 15°S/15°N and 30°S/30°N instead of at the equator. Niemeier and Schmidt (2017) calculated a smaller impact on the QBO for injections along a band between 30°N and 30°S. However, Tilmes et al. (2018) showed that also these different injection strategies impact the transport of species, e.g. ozone, due to different wave propagation in the stratosphere.

The impact of equatorial $SO_2$ injections on the QBO is qualitatively, but not quantitatively consistent across the different studies described above. Aquila et al. (2014) shows that the QBO vanishes at 2.5 $Tg(S)yr^{-1}$, and Niemeier and Schmidt (2017) at 8 $Tg(S)yr^{-1}$ equatorial injections. Jones et al. (2016) showed still oscillating winds with an injection of 7 $Tg(S)yr^{-1}$ and in Kleinschmitt et al. (2017) the QBO vanishes without developing a westerly jet as in the other models. Richter et al. (2017) showed a disappearance of the QBO with injections of 6 $Tg(S)yr^{-1}$ only when using prescribed chemistry. When using a fully interactive chemical module the QBO slows down, but does not disappear, at this equatorial injection rate. They related this to the additional heating and partly opposing cooling due to interactive ozone.

In this study we aim to understand the reasons behind the differences in the QBO response to $SO_2$ injections between two models, WACCM-110L and MAECHAM5-HAM. As none of the studies named above had the same simulation set-up, we





perform here simulations with WACCM-110L and MAECHAM5-HAM with the same set-up, $SO_2$ injection rate, and location. We describe the models and the performed simulations in Section 2, discuss the causes of the differences in Section 3, show in Section 3.5 that the models behave more similar when the amplitude of the aerosol heating is similar, and discuss shortly the impact of different horizontal resolution on the findings in Section 3.6. We end with a summary and discussion (Section 4).

## 5  2  Model Description and Simulations

### 2.1  MAECHAM-HAM

MAECHAM5-HAM, hereafter ECHAM, is a general circulation model (GCM), which is interactively coupled to an aerosol microphysical model. The simulations for this study were performed with the middle atmosphere version of the GCM ECHAM (Giorgetta et al., 2006). The horizontal resolution was about $2.8°$, spectral truncation at wave-number 42 (T42), and 90 vertical

layers up to $0.01\,\mathrm{hPa}$. ECHAM5 solves prognostic equations for temperature, surface pressure, vorticity, divergence, and phases of water. The vertical resolution allows the internal generation of the QBO in the tropical stratosphere (Giorgetta et al., 2006).

The prognostic modal aerosol microphysical model in ECHAM is HAM (Stier et al., 2005), which calculates the sulfate aerosol formation including nucleation, accumulation, condensation and coagulation, as well as its removal processes by sedi-

mentation and deposition. A simple stratospheric sulfur chemistry is applied above the tropopause (Timmreck, 2001; Hommel et al., 2011). The radiative direct effect of sulfate is included for both, SW and LW radiation, and coupled to the radiation scheme of ECHAM. The sulfate aerosol influences dynamical processes via temperature changes caused by scattering of short-wave radiation and absorption of near-infrared and longwave radiation. Within this stratospheric HAM version apart from the injected $SO_2$, only natural sulfur emissions are taken into account. These simulations use the model setup described in

Niemeier et al. (2009) and Niemeier and Timmreck (2015). The sea surface temperature (SST) is set to a climatological value (Hurrell et al., 2008), averaged over the AMIP period 1950 to 2000, and does not change due to CE.

### 2.2  WACCM-110L

WACCM-110L, hereafter WACCM, is a 'high top' version of the atmospheric component of the Community Earth System Model, version 1 (CESM1; Hurrell et al. (2013)) with 110 vertical levels instead of the default 70-levels. WACCM with 110

levels was developed for the SPARC QBO Initiative (QBOi, Butchart et al. (2018)) and this configuration of the model is in detail described in Garcia and Richter (2019). The horizontal resolution is $0.95°$latitude $\times$ $1.25°$longitude. The tropospheric physics and parameterizations in WACCM are exactly the same as in the Community Atmosphere Model, version 5 (CAM5), as well as updated physical parameterizations for planetary boundary layer turbulence, cloud microphysics, and aerosols, which has been described in detail by Mills et al. (2017). The gravity wave parameterization in the 110-level version of WACCM has

been adjusted to reproduce the observed period and amplitude of the QBO, as well as to produce extratropical stratospheric climate that's close to observed (see Garcia and Richter (2019) for full details).



Here, we use the specified chemistry version of WACCM, which uses a monthly varying present-day climatology to prescribe ozone, oxidants, and background stratospheric aerosols. Tropospheric aerosols are prognostically derived using the modal aerosol model (MAM3) (Liu et al., 2012). Direct effects and indirect effects of radiative effects of aerosols are included. Additionally, geoengineering sulfur injections into the stratosphere are performed similarly to ECHAM. As described in Mills

et al. (2016), the lack of interactive stratospheric chemistry, prevents OH values from depleting while reacting with the injected sulfur. This leads to a slightly faster formation of sulfate closer to the injection location, and with that a different lofting of aerosols in the tropics compared to a full chemistry version, as used in Mills et al. (2017). However, while the aerosol distribution is somewhat different than in the fully interactive chemistry version of WACCM, roughly 10% higher burden maximum in the tropics, the response of sulfur injections on the QBO is the same. The SST is prescribed and set to present day

values.

## 2.3  Simulations

The model simulations for this study follow the same protocol. Both models prescribe a repeating annual cycle of SSTs, present day, and chemical precursors, which are necessary for e.g. sulfur oxidation and radiative processes, on a monthly basis. These fields slightly differ between the two models but are not expected to have much influence on the simulation of the QBO. $SO_2$

was injected continuously over time into a single grid box at the equator at a height of 60 hPa with three different amounts of sulfur: 2, 4, 8 $Tg(S)yr^{-1}$. Simulations were carried out with ECHAM and WACCM for at least 20 years. Exact number of simulation years is shown in Table 1, as well as the number of years used to calculate time averages. ECHAM simulations were carried out longer, however no differences have been found between the results averaged over 20 years and the entire simulation length of ECHAM. Figures show either timeseries or zonal averages over time. Anomalies are calculated relative to

an average over a control run of 50 years (ECHAM) and 35 years (WACCM).

**Table 1.** Summary of simulations carried out with WACCM and ECHAM. Number of total simulation years is shown with the number of years in parentheses indicating the number of years used to calculate time averages of the different variables. The results are not sensitive to the number of years used to calculate time averages.

| Injection rate | WACCM | ECHAM (T42) | ECHAM (T63) |
|---|---|---|---|
| Control | 35 (35) years | 50 (50) years | 17 (15) |
| 2$Tg(S)yr^{-1}$ | 20 (15) | 17 (15) | |
| 4$Tg(S)yr^{-1}$ | 20 (15) | 50 (20) | 17 (7) |
| 8$Tg(S)yr^{-1}$ | 20 (15) | 40 (20) | 17 (7) |



# 3 Results

## 3.1 QBO Changes

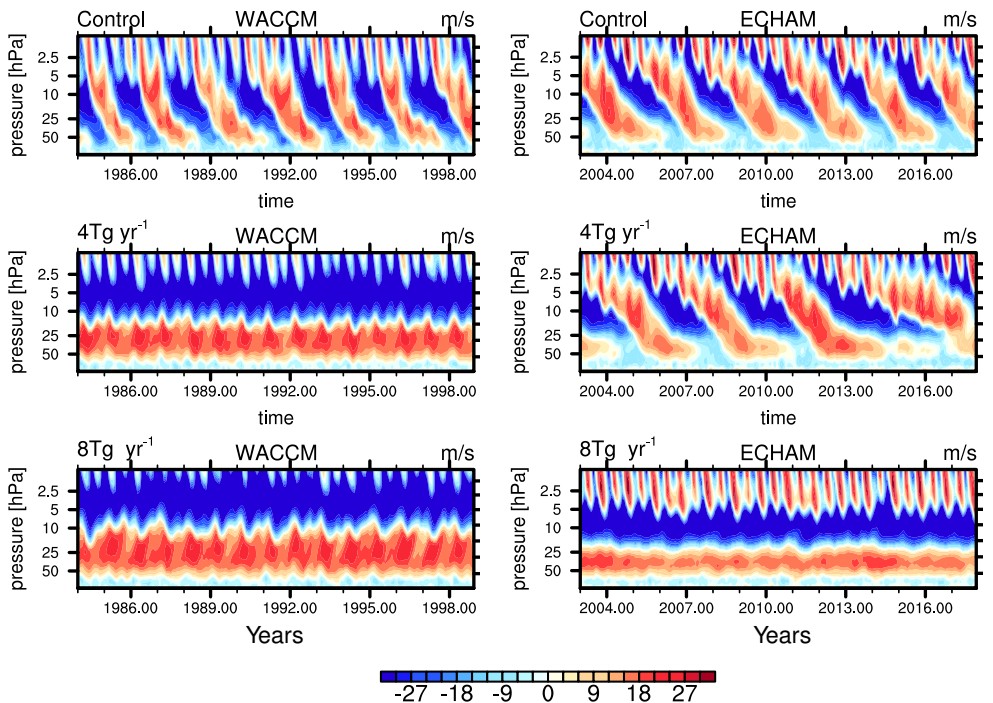

**Figure 1.** Zonal mean zonal wind $[\mathrm{ms}^{-1}]$ at the equator for a control simulation (top) and simulations with sulfur injections of $4\,\mathrm{Tg(S)\,yr}^{-1}$ (middle) and $8\,\mathrm{Tg(S)yr}^{-1}$ (bottom). Left: Results of WACCM. Right: Results of ECHAM.

Figure 1 shows the zonal mean zonal wind at the equator for the control simulation and two different injection rates for WACCM and ECHAM. Both models simulate the QBO well in the control simulation, without artificial injections of $SO_2$

5 (top). The QBO has an observed period of 28 month (Naujokat, 1986) on average. The simulated QBO period is about 27 months in WACCM and about 32 month in ECHAM. In WACCM the wind velocity is higher, slightly in the westerly phase but stronger in the easterly phase especially at altitudes below $20\,\mathrm{hPa}$, and the QBO propagates further down as in ECHAM. After the injection of sulfur into the tropical stratosphere the QBO responds quite differently to the same injection rate in the two models. While ECHAM shows a slower but still existing oscillation of the zonal wind for injections of $4\,\mathrm{Tg(S)yr}^{-1}$,

10 the oscillation of the zonal wind in WACCM completely vanishes, resulting in constant westerlies in the lower stratosphere, and easterlies above $\sim\!10\,\mathrm{hPa}$ (Fig 1, middle). Increasing the injection rate to $8\,\mathrm{Tg(S)yr}^{-1}$ increases slightly the velocity of the westerlies and the vertical extension of the westerly jet in WACCM (Fig 1, bottom). In ECHAM the oscillation vanishes at $8\,\mathrm{Tg(S)yr}^{-1}$ as well but wind velocity and vertical extension of the westerly jet are lower. The stronger westerly jets in



WACCM shifts the semi-annual oscillation (SAO) above 5 hPa to higher altitudes. For ECHAM the SAO still reaches 5 hPa for 8 Tg(S)yr$^{-1}$ injections gets shifted to higher altitudes also when the jets gets stronger with increasing injection rates (Niemeier and Schmidt, 2017). Thus, the QBO disappears in both models as a result of SO$_2$ injections, but at different injection rates.

## 3.2 Temperature and heating rate changes

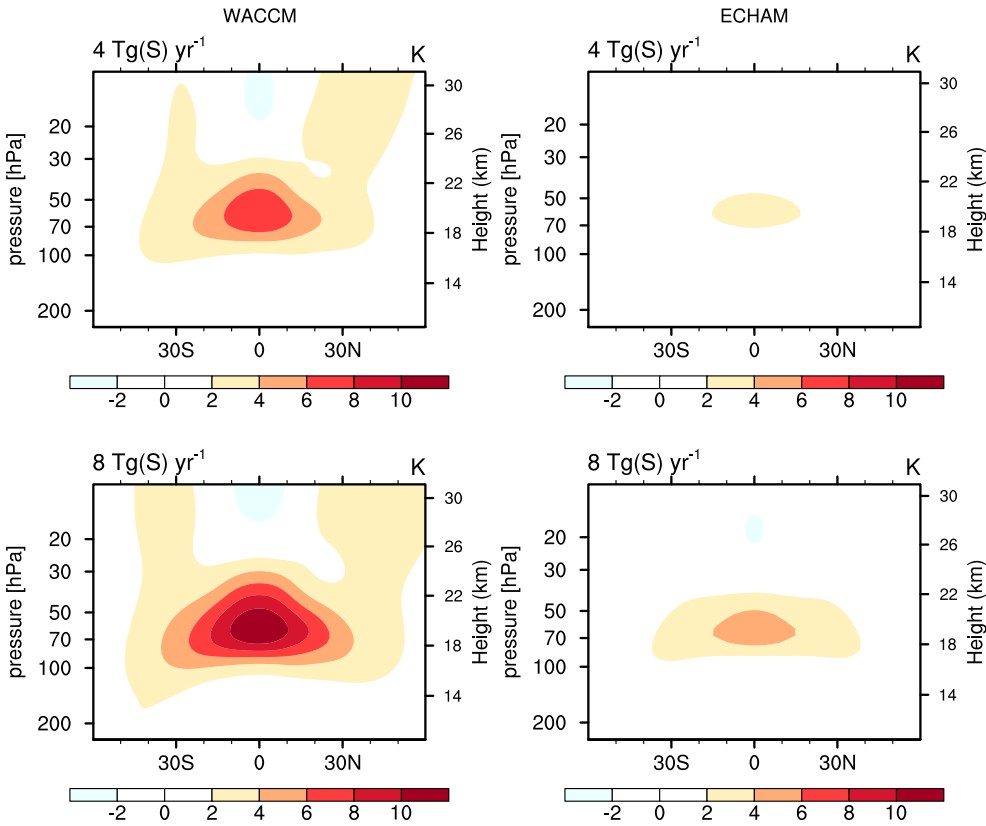

**Figure 2.** Temperature anomaly [K] caused by injections of 4 Tg(S)yr$^{-1}$ (top) and 8 Tg(S)yr$^{-1}$ (bottom) sulfur. Left: Results of WACCM. Right: Results of ECHAM5-HAM.

5     Aquila et al. (2014) identified the absorption of radiation by sulfate aerosols and the consequent heating in the lower strato-sphere as the main cause for the changes in the QBO. The heated sulfate layer impacts the thermal wind balance and vertical advection. This heating differs clearly between WACCM and ECHAM as can be seen in the amplitude of temperature anomalies in the stratosphere for both models (Figure 2). For the same sulfur injection rate, WACCM shows temperature anomaly roughly three times stronger than ECHAM for the 4 and 8 Tg(S)yr$^{-1}$ injections respectively. Therefore, the different response
10 of the QBO winds to the injection between the models is not surprising, as the thermal wind balance is much more strongly impacted in WACCM than in ECHAM.



**Figure 3.** Zonally averaged heating rates of near infrared (SW, top panels), terrestrial (LW, middle panels), and total (bottom panels) radiation of both models after an injection of 4 $\mathrm{Tg(S)yr^{-1}}$.

We investigate the reason for the different stratospheric heating in WACCM and ECHAM by examining the shortwave (SW) and terrestrial or longwave (LW) heating rates in both models for the simulation with 4 $\mathrm{Tg(S)yr^{-1}}$ injection. Similar results are found for the 8 $\mathrm{Tg(S)yr^{-1}}$ simulation, and hence not shown. Splitting of the heating rates into SW and LW components shows that SW heating rates for both models are of comparable amplitude (Figure 3, top panels) whereas there is a clearly higher heating rate for LW radiation than for near infrared (short wave, SW) radiation in WACCM (Figure 3). The heating





rates show that WACCM absorbs more than twice as much in the LW than in SW, while absorption is similar in between LW and SW in ECHAM. Compared to ECHAM, the LW absorption of sulfate in WACCM is three times stronger. We can see different absorption rates in LW and SW between the models, however, the stronger LW heating rate in WACCM corresponds to the stronger temperature anomaly in WACCM. Both models use the same radiation scheme, hence the differences can not

be explained by the radiation scheme. Therefore, we can assume that these differences in the LW and SW absorption cannot be the reason for the different temperature anomalies.

### 3.3 Sulfate properties

The zonally averaged sulfate burden, the vertically integrated sulfate concentration, shows at all latitudes a higher burden in WACCM than in ECHAM for the injection rate of 4 $\mathrm{Tg(S)yr^{-1}}$ (Figure 4). WACCM shows a distinct peak at the equator

while in ECHAM the distribution is much more even with latitude and the secondary maxima in the extra-tropics are only slightly smaller than the tropical maximum. This three to four times larger tropical sulfate burden in WACCM explains the larger temperature anomaly in WACCM, as more sulfate aerosols can absorb more radiation.

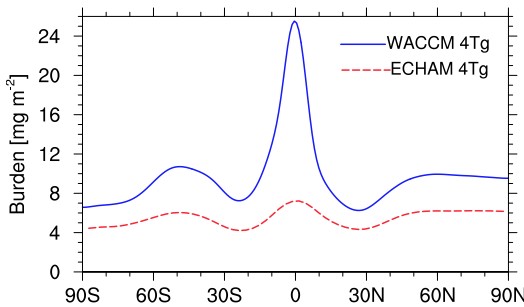

**Figure 4.** Zonally averaged sulfate burden of injections of 4 $\mathrm{Tg(S)yr^{-1}}$ for WACCM and ECHAM.

The vertical cross section of the zonally averaged sulfate concentrations reveals more details of the differences in distributions of sulfate in the two models (Figure 5). Not only the tropical concentration is higher in WACCM, in addition, the vertical

distribution of aerosols is very different between the two models. In ECHAM the sulfate is vertically advected to 25 hPa, while in WACCM sulfate reaches much higher altitudes and meridional transport mainly occurs mainly below 50 hPa. Vertical advection has to be much stronger in WACCM than in ECHAM to cause the differences. This is likely caused by a combination of a stronger lofting of aerosols as the result of radiative heating by aerosols, as well as resulting changes in the stratospheric wave propagation which causes an increase in the residual vertical velocity. Thus, from the comparison of the sulfur injection

cases only we cannot conclude whether a) the strong vertical advection is a consequence of the stronger heating or b) the cause of higher sulfate mass and, consequently, stronger heating. At this point we can only assume that the stronger tropical aerosol

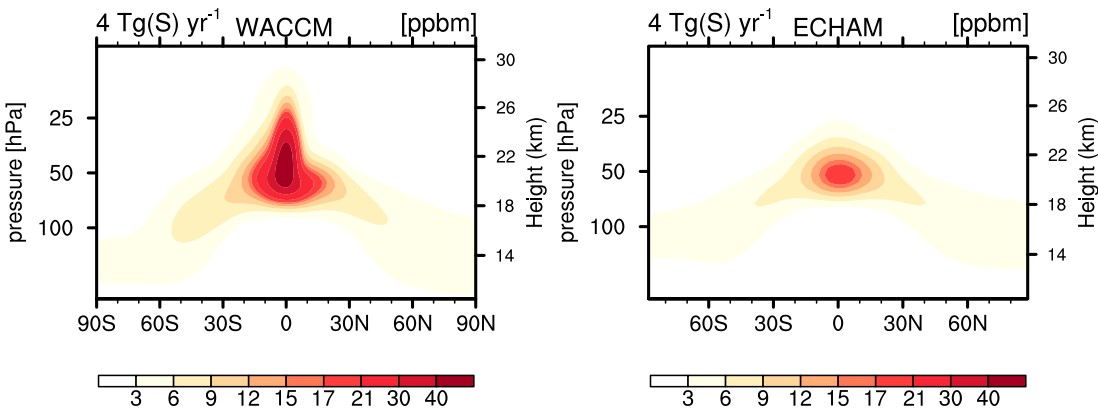

**Figure 5.** Zonally averaged sulfate concentration [ppbm] of injections of $4\,\mathrm{Tg(S)yr^{-1}}$ for WACCM and ECHAM.

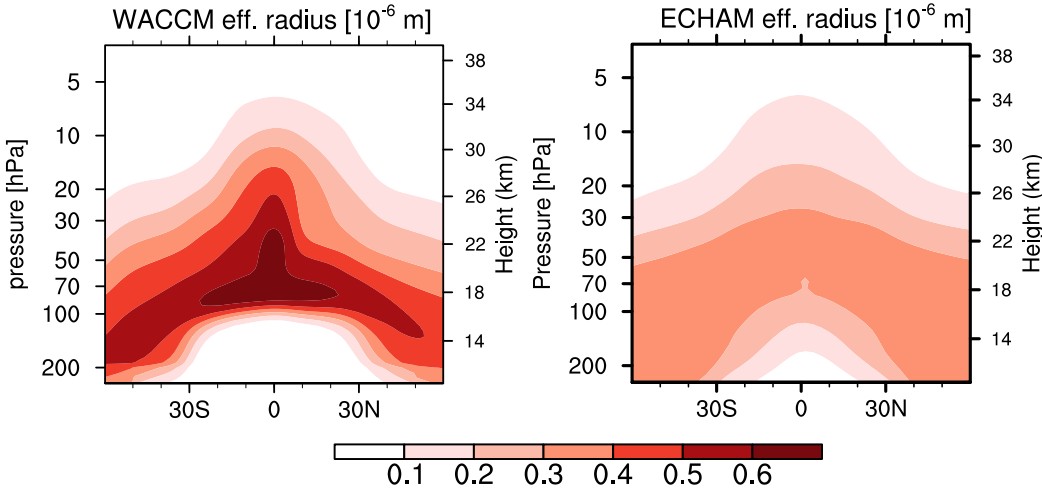

**Figure 6.** Effective radius [μm] for an injection rate of $4\,\mathrm{Tg(S)yr^{-1}}$ of WACCM (left) and ECHAM-HAM (right). WACCM shows only values in the stratosphere.

heating in WACCM is related to the higher sulfate load. The heating is a consequence of the sulfate burden, and not the source of the differences between the two models.

To further understand differences in the aerosol distribution and the resulting heating between WACCM and ECHAM, we examine the effective radii of aerosols in both models. This comparison (Figure 6) shows in the tropics twice as large radii for WACCM (0.6 μm) than for ECHAM (0.3 μm). The higher sulfur load results in larger particle radii, less scattering and, less SW radiative forcing (Dykema et al., 2016). LW radiative forcing depends on the sulfate mass and stays constant per injected





sulfur unit. The different radii are therefore not related to the differences in heating rates between the models. From the larger radii in WACCM we may assume a stronger sedimentation in the tropics in WACCM. But the burden is larger in WACCM. If sedimentation is a major difference between the models, the difference in the tropical burden between the two models should be smaller. An additional process, which determines the lifetime of the aerosols in the tropics is the vertical advection, the

5   residual vertical velocity $\omega^*$.

## 3.4   Dynamical changes

The patterns of the heating rates, sulfate concentrations and particle radii hint towards a stronger vertical advection in WACCM. A proxi for this behaviour is the residual vertical velocity, $\omega^*$. Richter et al. (2017) have shown that vertical advection plays a major role in dynamical changes in the tropical stratosphere. Visioni et al. (2018) showed a strong relation between the sulfate

10   lifetime and $\omega^*$. Therefore, we compare the residual vertical velocity of the control simulations (Figure 7) to get a more general impression of the behaviour of the two models, independently of additional updraft caused by the aerosol heating. In the altitude of the sulfur injection (60 hPa) and above shows WACCM an up to 70% stronger $\omega^*$ than ECHAM. This stronger $\omega^*$ results in a stronger vertical transport of the sulfate aerosols, which increases the tropical sulfate burden in WACCM. Additionally, the minimum of the $\omega^*$ profile is located at lower altitude in WACCM (70 hPa and 50 hPa in ECHAM), resulting in a stronger tropical confinement of the aerosols at this altitude.

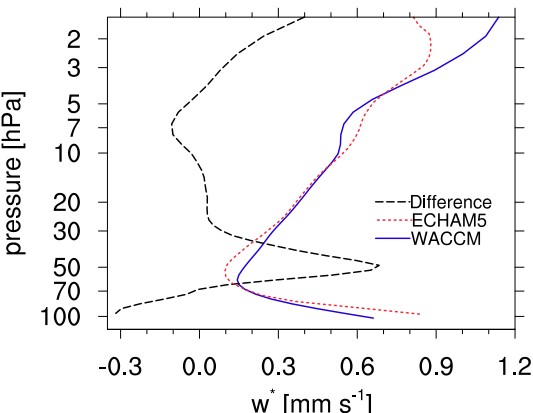

**Figure 7.** Residual vertical velocity of the control simulations in the tropics (averaged over 5°N to 5°S). Results of ECHAM (red) and WACCM (blue), and in black the difference (WACCM-ECHAM/ECHAM).

The consequence is twofold: a) a stronger $\omega^*$ counteracts more the downward propagation of the QBO shear zones and b) lifts the aerosols to higher altitudes, which increases the burden and, thus causes stronger heating. The heating of the aerosols further increases $\omega^*$, which shifts the minimum of $\omega^*$ downward (Figure 8). This can be seen in both models, but stronger in WACCM than in ECHAM. This feedback loop finally results in the vanishing of the QBO at an injection rate of 2 $\mathrm{Tg(S)yr}^{-1}$

20   in WACCM compared to 8 $\mathrm{Tg(S)yr}^{-1}$ in ECHAM. The reasons for the differences in $\omega^*$ in the control may lay in differences



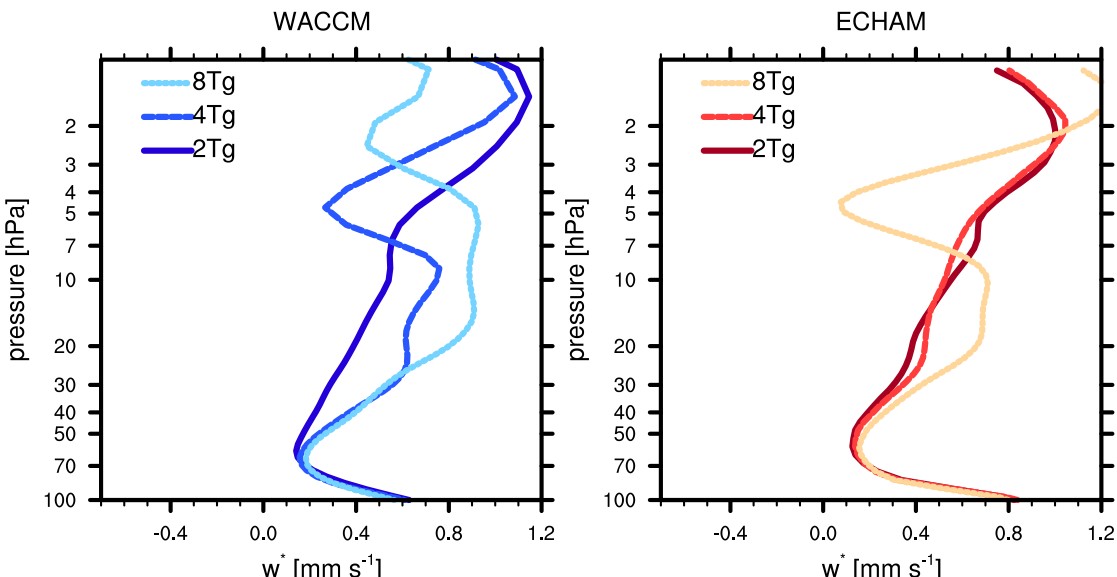

**Figure 8.** Residual vertical velocity in the tropics for injections of 2 Tg(S)yr$^{-1}$, 4 Tg(S)yr$^{-1}$ and 8 Tg(S)yr$^{-1}$ for WACCM (left) and ECHAM (right).

in the gravity wave parameterization and the relation how strongly resolved Rossby waves or parameterized gravity waves drive the upward mass flux (Cohen et al., 2014; SPARC, 2010). The better horizontal resolution in WACCM may also play a role. We conclude that the stronger $\omega^*$ in WACCM to be the main reason for the differences between the QBO response in the two models.

## 3.5 Comparison under the same heating conditions

Are differences in $\omega^*$ between the models the main cause of the difference in QBO impact or does the different heating also play an important role? To answer this question we compare different sulfur injection rates in the models that produce a similar heating rate in the sulfate layer. An injection rate of 2 Tg(S)yr$^{-1}$ in WACCM and 8 Tg(S)yr$^{-1}$ in ECHAM fulfills this criterium (Figure 9, top). Both experiments result in a temperature anomaly of $\sim$4 K in the tropical stratosphere. The heated area is slightly wider in ECHAM because the sulfate concentration (Figure 9, bottom) is slightly higher in the tropics and spreads more meridionally around 50 hPa. However, the maximum burden between the two models is rather similar in the tropics (Figure 10). The tropical maximum of the sulfate burden in ECHAM is only 2 mgm$^{-2}$ (12%) higher than in WACCM, despite a factor 4 higher injection rate of sulfur. The differences in the burden are larger in the extratropics ($\sim$50%). But the extratropical differences are not the focus of this study as we focus on the tropical stratosphere and the QBO.

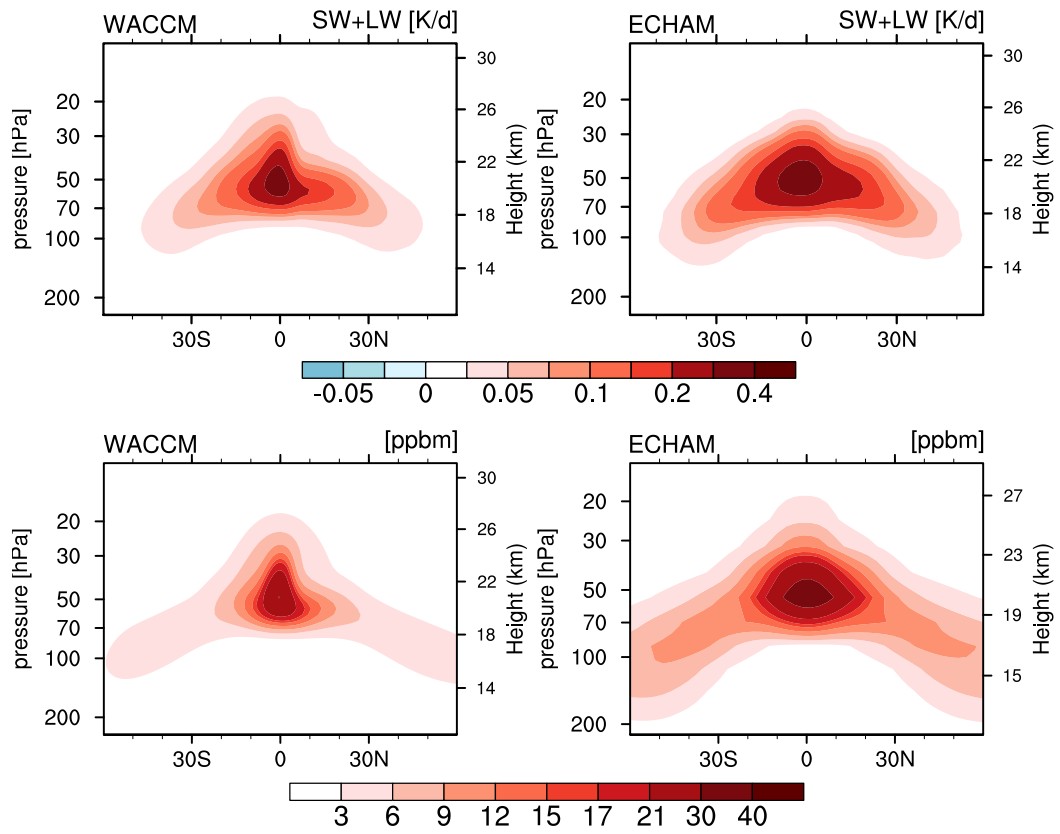

**Figure 9.** Top: Heating rate [K/d] and Bottom: sulfate concentration of a $2\,\mathrm{Tg(S)yr^{-1}}$ injection rate in WACCM (left) and an $8\,\mathrm{Tg(S)yr^{-1}}$ injection rate in ECHAM (right). Injection rates were chosen to result in similar temperature anomalies in both models.

The continuous westerly and easterly jets cause a different profile of $\omega^*$ than oscillating zonal winds under QBO conditions (Fig. 8). The clearly different profile of $\omega^*$ for the low injection cases to the $8\,\mathrm{Tg(S)yr^{-1}}$ in ECHAM and for all three injection

5   cases to the control in WACCM is a consequence of the disappearance of the QBO. We see a strong correlation of the pattern of the residual vertical velocity to the equatorial zonal wind profiles (Figure 1). The characteristic pattern of the vertical profile of $\omega^*$ in WACCM becomes similar in ECHAM-HAM when the oscillation of the equatorial jets vanishes at $8\,\mathrm{Tg(S)yr^{-1}}$ (Figure 8). Consequently, the maximum difference of $\omega^*$ between the models is only 30% when comparing the $2\,\mathrm{Tg(S)yr^{-1}}$ WACCM and the $8\,\mathrm{Tg(S)yr^{-1}}$ ECHAM injection cases (Figure 11). Differences occur mostly due to a vertical shift in the profiles. The

10  constant easterly and westerly jets cause distinct maxima and minima of $\omega^*$ below $50\,\mathrm{hPa}$. This compares well to the theory of the meridional and vertical transport processes within the QBO region and the secondary meridional oscillation (Plumb and Bell, 1982), which is caused by equatorward meridional advection in westerly jets and poleward within easterly jets combined with updraft in easterly shear and downdraft in westerly shear. E.g. the position of the maxima around 30 and $20\,\mathrm{hPa}$ are the





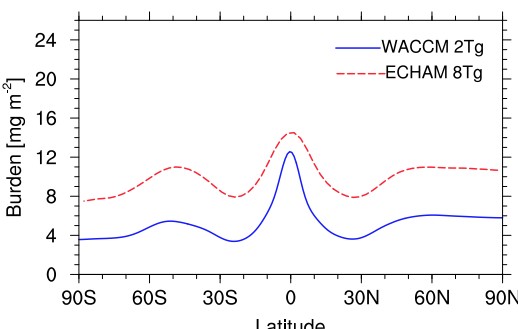

**Figure 10.** Zonally averaged sulfate burden of a 2 $Tg(S)yr^{-1}$ injection rate in WACCM (blue) and an 8 $Tg(S)yr^{-1}$ injection rate in ECHAM-HAM (red).

transition zones of the westerly and easterly jets. The heating of the aerosols, and the corresponding increased $\omega^*$, interferes with downdraft tendency in the westerly shear zone below 50 hPa. The result is the $\omega^*$ minimum around 60 hPa and the maximum between 30 hPa and 10 hPa agrees with the easterly shear zone (Figure 12).

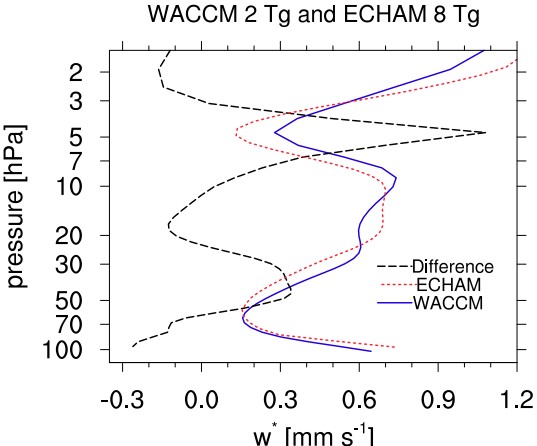

**Figure 11.** Figure 11: Residual vertical velocity in the tropics for a WACCM simulation with in injection rate of 2 $Tg(S)yr^{-1}$ (blue), ECHAM simulation with in injection rate of 8 $Tg(S)yr^{-1}$ (red), and the difference (black, (WACCM-ECHAM)/ECHAM).

Finally, we can say that the similar heating anomalies result in very similar zonal winds at the equator in both models (Figure 12). Both models show the vanishing of the QBO with a westerly jet in the lower stratosphere combined with an easterly jet at higher altitudes. WACCM simulates slightly higher wind velocities and less vertical extension of the westerly jet than ECHAM, but in general the response of the two models is very similar. Our findings suggest that the stronger tropical aerosol heating in





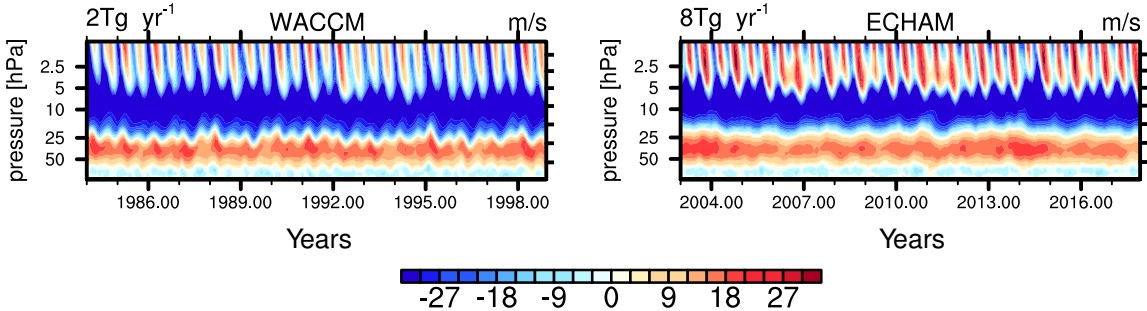

**Figure 12.** Zonal mean zonal wind at the equator of a 2 $\mathrm{Tg(S)yr}^{-1}$ injection rate in WACCM (right) and an 8 $\mathrm{Tg(S)yr}^{-1}$ injection rate in ECHAM.

WACCM is related to the higher sulfate load caused by the stronger $\omega^*$. The heating is a consequence of the sulfate burden, and not the source of the differences between the two models.

### 3.6 Impact of horizontal resolution

The horizontal resolution of the two models is very different in this study, $\sim 1°$ for WACCM, and $\sim 2.8°$ for ECHAM. Therefore, we examine the importance of different horizontal model resolutions on our results to reduce the number of uncertainties in our comparison. We increased the resolution of ECHAM from T42 to T63 ($\sim 1.8°$). This is still a coarser horizontal resolution than in WACCM but differences between the two simulations of ECHAM can indicate the impact of the horizontal resolution on the transport of sulfate out of the tropics and on the QBO.

When comparing the vertical velocity of T42 and T63 control simulations of ECHAM in the tropics (Fig. 13, left), we get a slight increase on $\omega^*$ (16%) in T63. This is much smaller than the difference to WACCM. Thus, we could expect a still smaller $\omega^*$ in ECHAM in case of a similar horizontal resolution. But we also see a slight shift of the minimum of $\omega^*$ to a lower altitude. From the differences to WACCM, the minimum of $\omega^*$ is at lower altitude in WACCM, we can expect a stronger confinement of the aerosols in the tropics. Indeed we find that the horizontal resolution has an impact on the simulated burden (Fig 13, right). The burden is about 30% higher in the tropics for injections of 8 TgS/yr in the T63 simulation as compared to the T42 simulation. The 4 $\mathrm{Tg(S)yr}^{-1}$ burden comes closer to the 2 $\mathrm{Tg(S)yr}^{-1}$ results of WACCM, with a substantial reduction of the peak in the aerosol burden in the tropics, but does not causes the QBO to vanish (not shown).

With T63 resolution the sulfate burden increases at all latitudes, not only in the tropics. The pattern of the burden indicates a slightly smaller residual meridional velocity and different isentropic mixing in the mid-latitudes in T63. Increasing the horizontal resolutions leads to a polar shift of the mid-latitude westerlies in the troposphere (Roeckner et al., 2006) with consequences of large scale wave propagation into the stratosphere. Additionally, Brühl et al. (2018) describe a better representation of sedimentation processes at high latitudes in T63. As we concentrate on the impact of sulfate on the QBO in this study, the differences in the extra-tropics will be left for further studies.





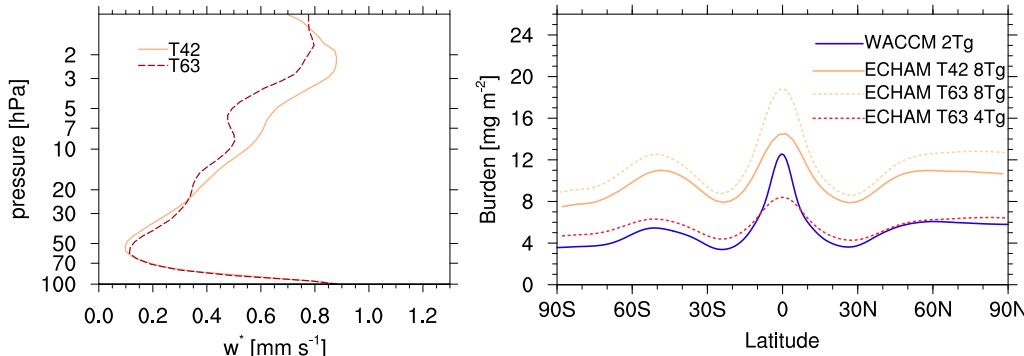

**Figure 13.** Left: Residual vertical velocity in the tropics of control simulations of ECHAM with T42 and T63 resolution. Right: Zonally averaged sulfate burden of WACCM (blue) and ECHAM T63 (dashed) and T42 (solid).

## 4   Summary and Discussion

We performed here simulations with different injection rates of $SO_2$ at the equator to compare the impact on the QBO in two
different general circulation models (WACCM and ECHAM). The QBO typically consists of alternating easterly and westerly zonal mean zonal winds, however in the presence of sulfur injections, the QBO sometimes vanishes, and turns into persistent westerlies in the lower stratosphere and persistent easterlies in the upper stratosphere. Both models used in the study had similar setup (e.g. prescribed SSTs and present day chemical precursors like OH or ozone) and were coupled to an aerosol microphysical model with three modes in WACCM and four modes in ECHAM. Both models qualitatively simulate an impact
on the QBO of sulfur injections similar to what was found in previous studies (Niemeier and Schmidt, 2017; Richter et al., 2017), however WACCM shows a disappearance of the QBO at an injection rate of 2 $Tg(S)yr^{-1}$ whereas ECHAM shows the disappearance of the QBO for an injection rate of 8 $Tg(S)yr^{-1}$.

We have shown that this difference results from different tropical vertical advection and different tropical residual vertical velocity, $\omega^*$, in the two models. $\omega^*$ differs not only in the simulations with $SO_2$ injections, but also in the control simulations
without any sulfur injection. $\omega^*$ is 70% larger than in ECHAM near the altitude of the $SO_2$ injection. Additionally, the minimum of $\omega^*$ is located at a lower altitude. At altitudes with a small $\omega^*$ meridional transport is enhanced, while a strong $\omega^*$ causes an enhanced tropical confinement of the aerosols. This confinement is stronger in WACCM above 50 $hPa$. Thus, the stronger $\omega^*$ results in a stronger vertical lifting, higher sulfate burden and, consequently, stronger heating of the stratosphere caused by aerosol absorption. This heating disturbs the thermal wind balance and causes an additional westerly momentum. Finally,
this results in the disappearance of the QBO at lower $SO_2$ injection rates than in ECHAM. This result partly opposes the assumptions of Kleinschmitt et al. (2017), who assumed the heating as main cause for different vertical advection in two models. It would be interesting to compare our results to the $\omega^*$ of their control simulation.

The reason for the different $\omega^*$ in the two models is complex. $\omega^*$, or the speed of the upwelling in the Brewer Dobson circulation is driven by a combination of larger scale (Rossby and synoptic-scale waves) and parameterized waves. The propagation



of waves and deposition of wave momentum by larger scale waves is impacted by numerous aspects of the model such as horizontal and vertical resolution, diffusion parameterization, physics parameterizations, which all differ between WACCM and ECHAM. Gravity wave parameterization contributions to driving the Brewer-Dobson circulation also vary between models (Butchart et al., 2011). WACCM and ECHAM have very different gravity wave parameterizations. It would be very difficult hence to isolate the reason for the different $\omega^*$ between WACCM and ECHAM, but simulations with different horizontal res-

olution shown in Section 3.6 have shown that horizontal resolution difference between WACCM and ECHAM contributed to the differences in $\omega^*$. Additionally, sedimentation may differ between the models as might be concluded from the difference in deposition when simulation a Tambora like volcanic eruption (Marshall et al., 2018). Sedimentation is a very important sink process for aerosols, especially at the poles, but three dimensional fields of sedimentation velocities were not available for both models. As we concentrated on the tropical stratosphere only, we leave this topic for further studies.

Finally we conclude that the difference in tropical upwelling even under present climate conditions between two models has a major impact on the projected effects of $SO_2$ injections on the QBO in WACCM and ECHAM. This is worrisome in terms of level of certainty of effects of $SO_2$ injections on stratospheric circulation in future climates, especially as the changes in the Brewer-Dobson circulation are uncertain, and in addition changes in gravity waves, which are a big driver of the QBO are even more uncertain in changing climate. Hence, a lot more research is needed before agreement is reached on how $SO_2$ injections

could affect the QBO. The reasons for the differences in this variable are too complex to give a recipe for a better agreement of the results.

*Code and data availability.* Primary data and scripts used in the analysis and other supplementary information that may be useful in reproducing the author's work are archived by the Max Planck Institute for Meteorology and can be obtained by contacting publications@mpimet.mpg.de. Model results of ECHAM are available under:

https://cera-www.dkrz.de/WDCC/ui/cerasearch/entry?acronym=DKRZ_LTA_550_ds00002 Model results of WACCM, and partly ECHAM, will also be made available via the cera databank of DKRZ.

*Author contributions.* All authors discussed the idea of the study and the setup of the models. UN performed ECHAM simulations, JR and ST WACCM simulations. UN and JR did the model comparison. UN wrote the manuscript with strong participation of JR and ST.

*Competing interests.* We declare no competing interests





*Acknowledgements.* We thank Andrea Segschneider for their helpful comments. This work is a contribution to the German DFG-funded Priority Program 'Climate Engineering: Risks, Challenges, Opportunities?' (SPP 1689). UN is supported by the SPP 1689 within the project CEIBRAL and CELARIT and DFG Research Unit VollImpact FOR2820 sub project TI344/2-1 UN got support from SPP 1689 and NCAR

5   for an scientific exchange in Boulder in 2016, where we discussed and stared the model comparison. The simulations were performed on the computer of the Deutsches Klima Rechenzentrum (DKRZ).

This work was supported by the National Center for Atmospheric Research, which is a major facility sponsored by the National Science Foundation under Cooperative Agreement No. 1852977. WACCM is a component of the Community Earth System Model (CESM), which is supported by NSF and the Office of Science of the U.S. Department of Energy. Computing resources were provided by NCAR's Climate

10  Simulation Laboratory, sponsored by NSF and other agencies. This research was enabled by the computational and storage resources of NCAR's Computational and Information Systems Laboratory (CISL). All simulations were carried out on the Yellowstone high-performance computing platform (CISL 2012).

CISL, 2012: Yellowstone: IBM iDataPlex /FDR-LIB. Computational and Information Systems Laboratory, http://n2t.net/ark:/85065/ d7FD3xhc.





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
