# Peer review of "Differing responses of the QBO to artificial $SO_2$ injections in two global models"

_Atmospheric Chemistry and Physics, 2020_

## Referee Comment (RC1) · Anonymous Referee #1 · 14 Apr 2020

General Comments

This is a nice study of one of the problems associated with stratospheric geoengineering via SO2 injection, namely how certain aspects of stratospheric dynamics are affected. Overall the paper is well written, though as indicated below there are a few points where more clarity is needed.

Specific Comments

Page 4, line 15: An approximate altitude for 60 hPa would be useful.

Page 4, line 17 (and Table 1): Which years are used to calculate the time averages - presumably the final "n" years in each case?

[Figure]

Interactive
comment

Page 6, line 2: This line makes no sense to me.

Page 8, lines 1-6: You show that the LW heating rate is larger in WACCM than in ECHAM but then say that this difference is not the cause of the larger temperature anomaly in WACCM. I think I understand what you're saying - that because the radiation codes are the same, the difference in LW heating rate must come from the inputs to the radiation scheme, but it still sounds odd. It's like saying "this heats up more but that's not why it's warmer". I suggest a re-write.

Page 10, lines 19-20: You say that the QBO vanishes in WACCM at an injection rate of 2 Tg[S]/yr but Figure 8 shows the vertical profile of omega_star in WACCM for 2 Tg to be very similar to the 2 and 4 Tg profiles in ECHAM, which you imply further on (Page 12, paragraph beginning line 3) are profiles characteristic of the presence of a QBO (which doesn't disappear in ECHAM unti 8 Tg). So if the WACCM 2 Tg omega_star profile looks like the ECHAM 2 & 4 Tg profiles which still have a QBO, why is the QBO absent at 2 Tg in WACCM despite it having a similar omega_star profile?

Page 12, lines 3-5 and Figure 8: In trying to see the impact of the absence of the QBO on the profile of omega_star it would be helpful to have the profiles from the Control simulations also plotted (perhaps in black) on Figure 8.

Page 14, line 11: "...a slight increase on omega_star" - only from 60 to 30 hPa: above that T63 has smaller values.

Technical Corrections

Page 9, Figure 5 (and Fig. 9[lower]): the intervals on the colour scale are very odd: first the interval is 3 ppb (up to 15 ppb), then it reduces to 2 ppb (to 17 ppb), then increase to 9 ppb and finally to 10 ppb. I've no problem with intervals which gradually change but I don't think one should use intervals which begin being constant, then decrease and then increase.

Page 10, line 8: "proxy" not "proxi".

Page 10, fig. 7: A vertical line to differentiate positive from negative values would be helpful.

Page 10, line 20: "lay" not "lie".

Page 15, line 15: Insert "in WACCM" before "is 70% larger".

Page 16, line 12: "simulating" not "simulation".

---

## Referee Comment (RC2) · Anonymous Referee #2 · 2 Jun 2020

General comments

This is an interesting study that explores reasons for different QBO responses to sulfate geoengineering between two models. The paper is very well written with a clear and thorough exploration of the differences. Please see below for minor comments and clarifications.

Although the models have a somewhat similar number of vertical levels (90 vs. 110), is there any major difference in the vertical resolution of the models in the stratosphere and do your results depend on the vertical resolution of the models? It would be useful to know if/how the vertical levels differ between the models and how, in addition to the horizontal resolution, the vertical resolution may contribute to the differences. Do you expect your results to be sensitive to the altitude of the injection?

[Figure]

It would be clearer if 'geoengineering' was included in the title, to avoid confusion with a volcanic SO2 injection pulse. It would also be useful to add a final sentence to the abstract that highlights the implications of this study.

Specific comments

P2L5: 'higher SO2 injections' – what sort of magnitude does this refer to?

P2L15: I do not quite follow by what you mean by 'in the steady state' in this context. Could you clarify?

P3L7: Please include the full model names. What is the height to which the WACCM levels reach?

P4L5: You mention that OH values in WACCM are not depleted – I understand this is also the case in ECHAM, so would be useful to mention here also. It would be helpful in the methods to explicitly state why you have chosen to use specified chemistry.

L4L15: Can you add the approximate altitude of the 60 hPa injection? Why was this height chosen?

Table1: Please consider only reporting the number of years used for the averages or state why not all years were taken. It would also be interesting to know how long it takes for the QBO changes to occur – does this differ between the models? Why was a 2 Tg experiment for ECHAM (T63) not included?

P6L9: It would be useful to state the magnitude of the heating

P8L2-4: I did not fully follow these sentences and the use of 'however'. Why are the different heating rates not a reason for different temperature anomalies?

P8L10: What causes this secondary maxima?

P10L1: How would different radii be related to differences in the heating rates? Do the different aerosol microphysical schemes between the models play a role? The different

modes are mentioned in section 4 but I think would be better placed in the methods.

Figure 8: It would be useful to show the control line to compare. How significant are these changes?

P11L14 (and P14L23): Could the extratropical differences in sulfate burden not also contribute to differences in the tropical QBO response? Can you comment on this?

P12L8: Where does the 30% maximum occur? It looks like the largest difference is much higher.

P14L1: It follows that the heating is a consequence of the sulfate burden, but it seems unclear to me by what you mean that it is not a source of the differences since the heating goes on to cause the QBO disruption.

P14L11: It appears the increase is below 30 hPa but not above. Could you clarify which altitudes you are referring to and why? It would also be useful to comment on the overall implications of the differing horizontal resolution results at the end of this section.

P16L20: The study has highlighted that the reasons for the differences in vertical advection are too complex to isolate in this study, but can some recommendation be given for future work to investigate this?

Technical corrections

P2L4: affects –> affecting

P2L14: direct –> directly, indirect –> indirectly

P2L26: 'also' would be better placed after 'strategies'

P2L28: shows –> showed. QBO vanishes at –> QBO vanishes with a 2.5 equatorial injection

P3L20: value –> values

P4L3: effects of effects. Can this be rephrased?

P4L4: 'additionally' can be removed

P5L5: month –> months

P5L7: as in –> than?

P6L2: missing word after injections

P8L16: two instances of 'mainly'

P13L5: agrees –> aligns?

P16L12: simulation –> simulating

---

## Author Comment (AC1) · 19 Jun 2020

**Answers to reviewers on the ACPD paper (acp-2020-206): Differing responses of the QBO to SO2 injections in two global models**

Ulrike Niemeier, Jadwiga H. Richter and Simone Tilmes

Max Planck Institute for Meteorology, Bundesstr. 53, 20146 Hamburg, Germany

We thank two anonymous reviewer for their helpful comments. We considered the recommendations carefully and made some changes in the text. The questions are in

bold, answers in italic and changes in the text in blue.

The questions of the reviewers made us aware of an error in Figure 8, where the legend was not correct. We changed Figure 8, and changed the interval of the color scale in Fig 5 and Fig 9.

We mad some changes in the text. E.g. we added some general words about the models and the performed experiments to the model description.

The titel has been changed to: Differing responses of the QBO to artificial SO2 injections in two global models

**Answers to reviewer1**

Thank you for the kind words. Your comment related to the discussion of Figure 8 made us aware of a wrong legend in the figure. This was very helpful.

**An approximate altitude for 60 hPa would be useful.**

60 hPa is about 19 km. We added in the text: of 60 hPa (about 19 km) with three different amounts of sulfur: 2, 4, 8 Tg(S) yr-1. An injection altitude of 6 hPa has been used in many previous studies (Niemeier and Timmreck, 2015; Tilmes et al., 2018). This is the upper range that can be reached with currently available planes.

Page 4, line 17 (and Table 1): Which years are used to calculate the time averages -presumably the final "n" years in each case?

Yes. We changed the table and show the number of yeares used for the time average only.

Page 6, line 2: This line makes no sense to me.

Thank you, a word was missing. We added: For ECHAM the SAO still reaches 5 hPa

for 8 Tg(S) yr-1 injections, but gets shifted to higher altitudes, similar to WACCM, when the jets gets stronger with increasing injection rates (Niemeier and Schmidt, 2017).

Page 8, lines 1-6: You show that the LW heating rate is larger in WACCM than in ECHAM but then say that this difference is not the cause of the larger temperature anomaly in WACCM. I think I understand what you're saying - that because the radiation codes are the same, the difference in LW heating rate must come from the inputs to the radiation scheme, but it still sounds odd. It's like saying "this heats up more but that'snot why it's warmer". I suggest a re-write.

*We rewrote the end of the paragraph:* In total (SW + LW), WACCM absorbs more than twice as much radiation than ECHAM. The stronger heating rate in WACCM corresponds to the stronger temperature anomaly in WACCM. Both models use the same radiation scheme, hence the differences can not be explained by the radiation scheme and must be caused by other processes in the model. E.g. the heating rate due to absorption of LS radiation depends on the sulfur mass.

Page 10, lines 19-20: You say that the QBO vanishes in WACCM at an injection rate of 2 Tg(S)/yr but Figure 8 shows the vertical profile of  $\omega^*$  in WACCM for 2 Tg to be very similar to the 2 and 4 Tg profiles in ECHAM, which you imply further on (Page12, paragraph beginning line 3) are profiles characteristic of the presence of a QBO (which doesn't disappear in ECHAM until 8 Tg). So if the WACCM 2 Tg  $\omega^*$  profile looks like the ECHAM 2 and 4 Tg profiles which still have a QBO, why is the QBO absent at 2 Tg in WACCM despite it having a similar  $\omega^*$  profile?

Thank you for mentioning the problem. Your comment is absolutely correct. The legend in Figure 8 was not correct. The figure showed 'CONTROL, 2Tg, 4Tg". In this combination the text made no sense. We corrected the legend and added the results of the 8 Tg simulation. Now the text describes the figure better and our findings can be see in the figure.

Page 12, lines 3-5 and Figure 8: In trying to see the impact of the absence of the

СЗ

QBO on the profile of  $\omega^*$  it would be helpful to have the profiles from the Control simulations also plotted (perhaps in black) on Figure 8.

See our comment above.

Page 14, line 11: "...a slight increase on  $\omega^*$ " - only from 60 to 30 hPa: above that T63 has smaller values.

Yes. 60 to 30 hPa is the altitude of the sulfate layer. We added to the text: When comparing the vertical velocity of T42 and T63 control simulations of ECHAM in the tropics (Fig. 13, left), we get a slight increase on  $\omega^*$  (16%) in T63 in the area of the sulfate layer around an altitude of 50 hPa.

Page 9, Figure 5 (and Fig. 9[lower]): the intervals on the colour scale are very odd: first the interval is 3 ppb (up to 15 ppb), then it reduces to 2 ppb (to 17 ppb), then increase to 9 ppb and finally to 10 ppb. I've no problem with intervals which gradually change but I don't think one should use intervals which begin being constant, then decrease and then increase.

Yes, we agree. Thank you for mentioning. There was a typo in the script. We changed the intervals.

Page 10, line 8: "proxy" not "proxi". Done

Page 10, fig. 7: A vertical line to differentiate positive from negative values would be helpful. Done

Page 10, line 20: "lay" not "lie. Done

Page 15, line 15: Insert "in WACCM" before "is 70% larger". Done

Page 16, line 12: "simulating" not "simulation". Done

**Answers to reviewer 2**

Thank you very much for the kind word and reviewing the paper. We went through all comments and ried to answer your questions carefully.

Although the models have a somewhat similar number of vertical levels (90 vs. 110), is there any major difference in the vertical resolution of the models in the stratosphere and do your results depend on the vertical resolution of the models? It would be useful to know if/how the vertical levels differ between the models.

WACCM has 32 levels and ECHAM 27 levels between 100 hPa and 10 hPa, the main area of interest of this study. The added plot (Fig. 1) shows the altitude of the vertical levels of both models. The spacing is regular in the relevant altitude (100 to 10 hPa). Previous studies have shown an increase of  $\omega^*$  when increasing the number of vertical levels in ECHAM (Niemeier and Tilmes, 2017), but the simulation with lower resolution does not generate an QBO in this paper. Thus the main differences can be related to the impact of the QBO.

We added to the text in Section 3.6: The vertical resolution of the models differs as well: 110 levels in WACCM and 90 levels in ECHAM. Within the area of interest, the sulfate layer between 100 and 10 hPa, the number of model levels differs by five grid points only (32 to 27 levels). This results roughly in a grid spacing of 0.5 km (WACCM) to 0.6 km (ECHAM) in this area. We do not expect a strong impact on  $\omega^*$  from this small difference in vertical resolution.

It would be clearer if 'geoengineering' was included in the title, to avoid confusion with a volcanic SO2 injection pulse. It would also be useful to add a final sentence to the abstract that highlights the implications of this study.

We added 'artificial' to the title and added to the abstract: The vertical velocity increases slightly in MAECHAM5-HAM when increasing the horizontal resolution. This study highlights the crucial role of dynamical processes and helps to understand the large uncertainties in the response of different models on artificial SO2 injections in climate engineering studies.

**How, in addition to the horizontal resolution, the vertical resolution may contribute to the differences.**

WACCM has five more levels in the area of interest. Following from the discussion above we assume the impact of this difference as small. We added text to Section 3.6.

**Do you expect your results to be sensitive to the altitude of the injection?**

Thank you for mentioning this aspect. As different previous studies show injections at different altitudes, the impact of an injection at another altitude should be mentioned.

The meridional transport of the sulfate aerosols out of the tropics depends on the altitude of the injection. The subtropical transport barrier is more permeable in the lower stratosphere. Therefore, the stratospheric confinement of the aerosols in the tropics is stronger when injecting at higher altitudes. This is also related to a higher value of the residual vertical velocity  $\omega^*$ .

Figure 7 shows that the differences of  $\omega^*$  in the control simulations of both models is largest around the injection altitude of 60 hPa. At higher altitudes, like 30 hPa, the injection would occur into an area with smaller differences in  $\omega^*$ . Thus we might expect smaller differences between the models. Niemeier and Tilmes (2017) and Tilmes et al. (2017) show results of zonally averaged sulfate burdens for two similar injection altitudes. The factor between the maxima, the higher injection altitude results in higher burden, is 2.2 in ECHAM and 1.7 in WACCM. Thus, we expect the results to be more similar when injecting at higher altitude.

*We added to the discussion:* In this study we compared injections at 6- hPa (about 19 km) only. This altitude shows the largest differences in  $\omega^*$  between the two models of

all altitudes. Therefore, injections at higher altitude, e.g. 30 hPa (24 km) would, most probably, cause less differences. Comparing results of Niemeier and Schmidt (2017) and Tilmes et al. (2018), both show results of injections at two altitudes, shows smaller differences between the models for the higher altitude injections.

**P2L5: 'higher SO2 injections' - what sort of magnitude does this refer to?**

We changed the text to:Model simulations have shown that under an artificial sulfate layer for tropical injections of 2 TgS  $yr^{-1}$  the QBO slows down, or even vanishes.

**P2L15: I do not quite follow by what you mean by 'in the steady state' in this context. Could you clarify?**

Yes, it sounds a bit odd. The sentence is better without these words.

**P3L7: Please include the full model names. What is the height to which the WACCM levels reach?**

ECHAM-HAM is name, not an abbreviation. We included the full name of WACCM in the text and added the altitude of 0.0006 hPa.

P4L5: You mention that OH values in WACCM are not depleted – I understand this is also the case in ECHAM, so would be useful to mention here also. It would be helpful in the methods to explicitly state why you have chosen to use specified chemistry.

We added some lines before the detailed descriptions of the models: The setup of the models aims on the best comparability of the results. Therefore, the single features of the models were chosen to be alike. E.g. one model has no interactive chemistry for precursors of SO2 oxidation. Therefore prescribe both models the precursors on a monthly mean basis. They prescribe also a repeating annual cycle of SSTs, present day. These fields slightly differ between the two models but are not expected to have much influence on the simulation of the QBO. Both models are coupled to modal aerosol microphysical models. The number of modes differs, nucleation, Aitken.

accumulation and, coarse mode in MAECHAM5-HAM and Aitken, accumulation and, coarse mode in WACCM.

**L4L15: Can you add the approximate altitude of the 60 hPa injection? Why was this height chosen?**

60 hPa is about 19 km. We added in the text: ...of 60 hPa (about 19 km) with three different amounts of sulfur: 2, 4, 8 Tg(S)  $yr^{-1}$ . An injection altitude of 60 hPa has been used in many previous studies (Niemeier and Timmreck, 2015; Tilmes et al., 2018, e.g.). This is the upper range that can be reached with currently available planes.

Table 1: Please consider only reporting the number of years used for the averages or state why not all years were taken. It would also be interesting to know how long it takes for the QBO changes to occur – does this differ between the models? Why was a 2(T63) not included?

We followed your recommendation and changed the table. A 2 Tg experiment for ECHAM was not available and is not used this study.

**P6L9: It would be useful to state the magnitude of the heating**

*We added to the text:* WACCM simulates maxima of temperature anomalies of 5.7 and 12.5 K for injections of 4 and 8 Tg(S)  $yr^{-1}$ , ECHAM only 2 and 4.5 K. Thus,

**P8 L2-4: I did not fully follow these sentences and the use of 'however'. Why are the different heating rates not a reason for different temperature anomalies?**

We changed the text: In total (SW + LW), WACCM absorbs more than twice as much radiation than ECHAM. The stronger heating rate in WACCM corresponds to the stronger temperature anomaly in WACCM. Both models use the same radiation scheme, hence the differences can not be explained by the radiation scheme and must be caused by other processes in the model. E.g. the heating rate due to absorption of LS radiation depends on the sulfur mass.

**P8L10: What causes this secondary maxima?**

Transport to the high latitudes is blocked in winter. The consequence are higher concentrations in mid-latitudes. We added to the text: WACCM shows a distinct peak at the equator while in ECHAM the distribution is much more even with latitude and the secondary maxima, caused by the blocking of meridional transport by the polar vortex in the winter hemisphere, in the extra-tropics are only slightly smaller than the tropical maximum.

P10L1: How would different radii be related to differences in the heating rates? Do the different aerosol microphysical schemes between the models play a role? The different modes are mentioned in section 4 but I think would be better placed in the methods.

You understood the sentence differently as it was supposed. We say in the sentence before that LW absorption is independent of the particle size. As a consequence, we cannot relate LW heating rate and particle size. To prevent a misunderstanding, we changed the text: LW radiative forcing depends on the sulfate mass and stays constant per injected sulfur unit and is not related to particle radii.

**Figure 8: It would be useful to show the control line to compare.**

The legend in Figure 8 was not correct. The figure showed 'CONTROL, 2Tg, 4Tg". We corrected the legend and added the results of the 8 Tg simulation. Now text and figure are fitting. We added a control line as well.

**P11L14 (and P14L23): Could the extra tropical differences in sulfate burden not also contribute to differences in the tropical QBO response? Can you comment on this?**

The main transport follows the Brewer-Dobson circulation in the stratosphere, rising air masses in the tropics and transport toward the poles. Exchange between the tropics and extra-tropics is hindered by the sub-tropical transport barrier. Even wave driven turbulent transport occurs against the BDC, transport into the tropics is small. Therefore, we do not expect that the differences in the extra-tropics can influence the situation in the tropics.

**P12L8: Where does the 30% maximum occur? It looks like the largest difference is much higher.**

This maximum occurs between 50 and 30 hPa. We changed the text to: Consequently, the maximum difference of  $\omega^*$  between the models, in the main area of the sulfate layer between 70 and 20 hPa, is only 34% when comparing the 2 Tg(S) yr-1 WACCM and the 8 Tg(S) yr-1 ECHAM injection cases (Figure 11).

P14L1: It follows that the heating is a consequence of the sulfate burden, but it seems unclear to me by what you mean that it is not a source of the differences since the heating goes on to cause the QBO disruption.

Yes, the heating is the source of the QBO disruption, but the differences between the models are caused by the different  $\omega^*$ . We changed the sentence to: Our findings suggest that the stronger tropical aerosol heating in WACCM is a consequence of the higher sulfate concentrations. The source of the differences between the two models, and the cause of the higher concentrations in WACCM, is the different  $\omega^*$ .

P14L11: It appears the increase is below 30 hPa but not above. Could you clarify which altitudes you are referring to and why? It would also be useful to comment on the overall implications of the differing horizontal resolution results at the end of this section.

We added the altitude of the area of interest to the text: When comparing the vertical velocity of T42 and T63 control simulations of ECHAM in the tropics (Fig. 13, left), we get a slight increase on  $\omega^*$  (16%) in T63 in the area of the sulfate layer around an altitude of 50 hPa.

We changed the last paragraph of this section to: Increasing the horizontal resolution

to T63 leads to a polar shift of the mid-latitude westerlies in the troposphere (Roeckner et al., 2006) with consequences on large scale wave propagation into the stratosphere. The dynamical changes result in an increase of With T63 resolution the sulfate burden increases at all latitudes, not only in the tropics. The pattern of the burden indicates a slightly smaller residual meridional velocity and different isentropic mixing in the mid-latitudes in T63. Additionally, Brühl et al. (2018) describe a better representation of sedimentation processes at high latitudes in T63. As we concentrate on the impact of sulfate on the QBO in this study, the differences in the extra-tropics will be left for further studies.

P16L20: The study has highlighted that the reasons for the differences in vertical advection are too complex to isolate in this study, but can some recommendation be given for future work to investigate this?

We added at the end of the conclusion: The model intercomparison initiative GeoMIP6 (Kravitz et al., 2015) has mainly concentrated on climate impact of CE. A simple SO2 injection experiment with well defined input parameters for GCM with aerosol microphysical modules, e.g. grid resolution, injection, model setups, may create further understanding on related differences and uncertainties.

P2L4: affects -> affecting Done P2L14: direct -> directly, indirect -> indirectly Done P2L26: 'also' would be better placed after 'strategies' Done P2L28: shows -> showed. QBO vanishes at -> QBO vanishes with a 2.5 equatorial injection Done

P3L20: value -> values Done P4L3: effects of effects. Can this be rephrased? P4L4: 'additionally' can be removed Done P5L5: month -> months Done

P5L7: as in -> than? Done
P6L2: missing word after injections Done, see above.
P8L16: two instances of 'mainly' Done
P13L5: agrees -> aligns? Done
16L12: simulation -> simulating Done

**References**

Brühl, C., Schallock, J., Klingmüller, K., Robert, C., Bingen, C., Clarisse, L., Heckel, A., North, P., and Rieger, L.: Stratospheric aerosol radiative forcing simulated by the chemistry climate model EMAC using Aerosol CCI satellite data, Atmospheric Chemistry and Physics, 18, 12845–12857, 10.5194/acp-18-12845-2018, 2018.

Butchart, N., Anstey, J. A., Hamilton, K., Osprey, S., McLandress, C., Bushell, A. C., Kawatani, Y., Kim, Y.-H., Lott, F., Scinocca, J., Stockdale, T. N., Andrews, M., Bellprat, O., Braesicke, P., Cagnazzo, C., Chen, C.-C., Chun, H.-Y., Dobrynin, M., Garcia, R. R., Garcia-Serrano, J., Gray, L. J., Holt, L., Kerzenmacher, T., Naoe, H., Pohlmann, H., Richter, J. H., Scaife, A. A., Schenzinger, V., Serva, F., Versick, S., Watanabe, S., Yoshida, K., and Yukimoto, S.: Overview of experiment design and comparison of models participating in phase 1 of the SPARC Quasi-Biennial Oscillation initiative (QBOi), Geoscientific Model Development, 11, 1009–1032, 10.5194/gmd-11-1009-2018, https://www.geosci-model-dev.net/11/1009/2018/, 2018.

Kravitz, B., Robock, A., Tilmes, S., Boucher, O., English, J. M., Irvine, P. J., Jones, A., Lawrence, M. G., MacCracken, M., Muri, H., Moore, J. C., Niemeier, U., Phipps, S. J., Sillmann, J., Storelvmo, T., Wang, H., and Watanabe, S.: The Geoengineering Model Intercomparison Project Phase 6 (GeoMIP6): Simulation design and preliminary results, Geoscientific Model Development, 8, 3379–3392, 10.5194/gmd-8-3379-2015, 2015.

Niemeier, U. and Schmidt, H.: Changing transport processes in the stratosphere by radiative heating of sulfate aerosols, Atmospheric Chemistry and Physics, 17, 14871–14886, 10.5194/acp-17-14871-2017, https://www.atmos-chem-phys.net/17/ 14871/2017/, 2017.

Niemeier, U. and Tilmes, S.: Sulfur injections for a cooler planet, Science, 357, 246–248, 10.1126/science.aan3317, http://science.sciencemag.org/content/357/6348/246, 2017.

Niemeier, U. and Timmreck, C.: What is the limit of climate engineering by stratospheric injection of SO2?, Atmospheric Chemistry and Physics, 15, 9129–9141, 10.5194/acp-15-9129-2015, http://www.atmos-chem-phys.net/15/9129/2015/, 2015.

Roeckner, E., Brokopf, R., Esch, M., Giorgetta, M., Hagemann, S., Kornblueh, L., Manzini, E., Schlese, U., and Schulzweida, U.: Sensitivity of simulated climate to horizontal and vertical resolution in the ECHAM5 atmosphere model, J. Climate, 19, 3771–3791, 2006.

Tilmes, S., Richter, J. H., Mills, M. J., Kravitz, B., MacMartin, D. G., Vitt, F., Tribbia, J. J., and Lamarque, J.-F.: Sensitivity of Aerosol Distribution and Climate Response to Stratospheric SO2 Injection Locations, Journal of Geophysical Research: Atmospheres, 122, 12,591–12,615, 10.1002/2017JD026888, https://agupubs.onlinelibrary.wiley.com/doi/abs/10.1002/2017JD026888, 2017.

Tilmes, S., Richter, J. H., Mills, M. J., Kravitz, B., MacMartin, D. G., Garcia, R. R., Kinnison, D. E., Lamarque, J.-F., Tribbia, J., and Vitt, F.: Effects of Different Stratospheric SO2 Injection Altitudes on Stratospheric Chemistry and Dynamics, Journal of Geophysical Research: Atmospheres, 123, 4654–4673, 10.1002/2017JD028146, https://agupubs.onlinelibrary.wiley.com/doi/abs/10.1002/2017JD028146, 2018. %bibliographyreferences

Fig. 1. Vertical levels of WACCM (orange, left) and ECHAM (red, right) between 100 and 10 hPa.